# Regulatory Mechanisms and Reversal of CD8^+^T Cell Exhaustion: A Literature Review

**DOI:** 10.3390/biology12040541

**Published:** 2023-04-01

**Authors:** Wanwan Zhu, Yiming Li, Mingwei Han, Jianli Jiang

**Affiliations:** National Translational Science Center for Molecular Medicine, Department of Cell Biology, Fourth Military Medical University, Xi’an 710000, China; zww09262021@163.com (W.Z.); lym052@fmmu.edu.cn (Y.L.); han212313@163.com (M.H.)

**Keywords:** CD8^+^T cells, exhaustion, inhibitory receptors, PD-1, immunotherapy

## Abstract

**Simple Summary:**

It has been known for decades that immune T cells, especially due to their capacity for powerful antigen-directed cytotoxicity, have become a central focus for engaging the immune system in the fight against cancer. Interest in harnessing T cells for cancer immunotherapy is rapidly growing. The latest clinical research aims to effectively use immune checkpoint inhibitor (ICI) drugs to treat cancer patients. However, T cells can either establish a protective antitumor response, or, in contrast, induce severe dysfunction that promotes disease progression in the majority of chronic infection patients. The contradiction depends, to a large degree, on the interaction between immune T cells and other tumor infiltrating cells or inflammatory cytokines around the tumor microenvironment. This review aims to summarize the current knowledge and regulatory mechanisms of CD8^+^T cell exhaustion, since reversing T cell exhaustion will definitely become a biomarker of clinical tumor immunotherapy.

**Abstract:**

CD8^+^T cell exhaustion is a state of T cell dysfunction during chronic infection and tumor progression. Exhausted CD8^+^T cells are characterized by low effector function, high expression of inhibitory receptors, unique metabolic patterns, and altered transcriptional profiles. Recently, advances in understanding and interfering with the regulatory mechanisms associated with T cell exhaustion in tumor immunotherapy have brought greater attention to the field. Therefore, we emphasize the typical features and related mechanisms of CD8^+^T cell exhaustion and particularly the potential for its reversal, which has clinical implications for immunotherapy.

## 1. Introduction

In general, with the developing period of tumor immunotherapy, research into T cell exhaustion has progressed rapidly. T cell exhaustion is described as a phenomenon of markedly low T-cell functional responses in patients with sustained HIV (human immunodeficiency virus) infection [1]. Researchers have, therefore, hypothesized that viral persistence is associated with T cells with impaired function [2]. Gallimore [3] and Zajac [4] discovered that persistent viral infection is closely associated with the low affinity of MHCI for antigenic peptides and reduces its role in T cell cytotoxicity. This was determined by applying a staining technique in which MHCI was used to track virus-specific CD8^+^T cells. Accordingly, the concept of CD8^+^T cell exhaustion was proposed, specifically referring to the dysfunction of specific subsets of T cells, including cytotoxic lymphocytes, during chronic viral infections [2,5]. It is commonly believed in the field of immunology that T cells (especially CD8^+^T cells) play an indispensable role in killing tumor cells. Recently, however, in some models, natural killer (NK) [6] cells show cytotoxic activity against diverse tumor cell types. This activity reduces the production of cytokines, such as interferon γ (IFNγ), and regulates adaptive antitumor immune responses, stemming a multitude of effects on the growth and maintenance of cancer [7]. Recently, it was reported that lymphocyte exhaustion is not only T cell exhaustion but also NK cell exhaustion [8]. It was reported that the multiple cell exhaustion, for instance the combination of T cell and NK cell exhaustion, may exist in several cancer patients [9,10].

Since then, it was proved that CD8^+^T cell exhaustion [5,11] exists in several animal models of chronic viral infection, mirroring the T cell exhaustion observed in multiple cancers, such as melanoma and breast cancer. Exhausted CD8^+^T (T_EX_) cells, consisting of heterogeneous cell populations with unique functional states, play important roles in cancer, chronic infections, and autoimmunity [12,13]. More detailed analyses of CD8^+^T cell exhaustion have since been demonstrated, and some experts in immunotherapy summarized and defined the state of T cell exhaustion in 2019 [14], further clarifying the concrete concept. All in all, we are concerned about the characteristics and mechanisms of CD8^+^T cell exhaustion and emphasize approaches to reverse it for clinical applications in tumor therapy.

## 2. The Features of T Cell Exhaustion

Exhausted T cells belong to a unique cell lineage composed mainly of heterogeneous cells, including progenitor and terminal subsets [14]. In contrast to memory T cells (T_MEM_) and effector T cells (T_EFF_), T_EX_ cells are distinguished by the features of progressive loss of effector functions (cytokine production and killing capacity), highly sustained inhibitory receptor expression, distinct metabolic patterns, and unique transcriptional and epigenetic features [14,15]. These concepts are discussed in further detail below and are illustrated in Figure A1. 

### 2.1. Progressive Loss of Effector Function

With regard to effector function, it is believed that T_EX_ cells show progressive dysfunction and even loss of function [16]; high proliferation, killing capacity, and interleukin-2 (IL-2) secretion are usually diminished [17]. Subsequently, secretion of some cytokines, such as tumor necrosis factor (TNF), is extremely reduced. Eventually T cells will partial or complete lose T killing capacity, eventually causing CD8^+^T cells to form a complex set of noncanonical hyporesponsive T_EFF_ subsets [18]. This phenomenon has been seen in a variety of mouse and human tumor models, including melanoma [19], chronic myeloid leukemia [20], ovarian cancer [21], and chronic lymphocytic leukemia (CLL) [22].

### 2.2. Sustained and High Expression of Inhibitory Receptor Molecules

One unique characteristic of T_EX_ cells during chronic infection or tumor immune response is the persistent upregulation and co-expression of various inhibitory receptors (IRs) [13], including programmed cell death protein 1 (PD-1). IRs [23] are concurrently present on CD8^+^T cells at high levels, negatively interfering with T cell activation and affecting T cell cytotoxic killing function. It is generally considered that the level of inhibitory receptor expression on CD8^+^T cells greatly affects the degree of CD8^+^T cell exhaustion. Without IR expression, CD8^+^T cells can effectively kill tumors and exert antiviral functions. Recent studies have found that inhibitory receptors execute their functions mainly via the following mechanisms [11,24]: (1) interfering with T cell receptor (TCR) activation by the downstream signaling pathways; or (2) upregulating gene markers related to CD8^+^T cell exhaustion.

#### 2.2.1. PD-1

Based on previous research, the first IRs to be identified was programmed cell death receptor-1 (PD-1), which are the family of the CD28 transmembrane protein receptors. PD-1 interacts specifically with PD-L1 to suppress anticancer T-cell immunity in multiple human cancers [25,26]. Numerous studies have identified additional mechanisms of PD-1 and PD-L1 activity. For example, PD-L1 triggers inhibition of T cell proliferation through tyrosine phosphorylation in the cytosolic region of TCRs, the mutual interaction therefore downregulates T cell activation [27]. Barber et al. [28], analyzed the differential gene expression in the virus-specific CD8^+^T cells during chronic infection and found that PD-1 largely up-regulating over-expressed in functionally T_EX_ cells [29,30,31]. Typically, PD-1 is transiently expressed in activated CD8^+^T cells following the acute viral infection and its level largely keep normal after a short period of viral antigen activation. However, in T_EX_ cells, PD-1 is constantly expressed at high levels [30,31,32]. Yokosuka et al. found that PD-1 can bind to TCRs navigates T cell exhaustion by recruiting SHP2 which are mainly active TCR signaling pathways in the opposing aspects. Recent studies have displayed that the blockade of PD-1 can change the exhaustion of the T cells and induce immune cells powerfully killing cancer cells [33]. Wei et al. [34] studied the levels of PD-1 in primary human T cells and examined T cell functions that are PD-1-sensitive (cytokine secretion, Ca^2+^flux, and proliferative capacity), which was achieved by stimulating the T cells using APC cells overexpressing PD-L1, demonstrated that high PD-1 expression of the T cells were far inferior to the low or no PD-1 expression in cytotoxic function. 

#### 2.2.2. LAG-3

Lymphocyte activation gene 3 (LAG-3) is highly expressed on exhausted T cells which is the family of the non-ITIM inhibitory receptor. LAG-3 mainly acts in negative regulating the T cell cycle via the KLEELE motif [35,36]. LAG-3 connects with its ligand, which will greatly suppress the T_EFF_ activity, promoting immune tolerance to the tumor microenvironment (TME). Antibodies against LAG-3 have been tested in repeated clinical trials, particularly in metastatic breast and pancreatic cancers; blocking LAG-3 alone had little effect on T_EX_ function during chronic LCMV infection, however, blocking LAG-3 in the combination of the IR of the PD-1 can forcefully regain T-cell cytotoxicity. Clinically, there have been many breakthroughs in LAG-3-targeted therapies in recent years; for example, the newly approved LAG-3 immune checkpoint inhibitor Relatlimab plus the PD-1 inhibitor Nivolumab [37,38] for melanoma treatment, can more than double the survival rate and reduce treatment side effects versus nivolumab alone. Furthermore, the combination also has a high clinical efficacy against leukemia [39]. Fortunately, in March 2022, nivolumab [40] in combination with relatlimab (Opdualag^TM^) received approval in the some areas such as USA for the treatment of unresectable melanoma [41].

#### 2.2.3. TIGIT

TIGIT is a type of IR-containing ITIM, whose ligands include CD155, CD112, and PVRL3. TIGIT competitively binds CD155/CD112 on the APC/tumor cells which exerts immunosuppressive effects on CD8^+^T cells that are highly expressed the multiple IRs, eventually resulting in the formation of exhausted T cells that have lost their tumor-killing effect [31,42,43]. TIGIT binding CD155 expressed by cancer cells results in impaired cytokine secretion by TIGIT^+^CD8^+^T cells, and their activation is inhibited. An anti-TIGIT candidate drug can closely engage to TIGIT and consequently hinder its interaction with CD155 [44]. Furthermore, blocking TIGIT and PD-1 has been shown more effective in restoring the function of T cells and improving the duration of survival of the mice [45]. In some clinical cancer patients, PD-1 is co-expressed in tumor-infiltrating CD8^+^T cells, and the combined blockade of TIGIT and the barrier in the PD-1/PD-L1 interactions can greatly enhance cytotoxic effect of CD8^+^T cells [46]. Hence, TIGIT has become a reliable therapeutic target for clinical tumor immunotherapy.

#### 2.2.4. VISTA

VISTA also known as PD-1H, is a type I transmembrane protein which is encoded by the human VSIR, and serves as a novel checkpoint target in immunotherapy [47]. It binds to two ligands, V-set and Ig domain-containing 3 (VSIG-3) and P-selectin glycoprotein ligand-1 (PSGL-1), which exert inhibitory T-cell activity and block effective antitumor responses [48,49]. As currently, some researches has showed that VISTA can render patients resistant to immunotherapy [49,50]; Kalakand et al. found that lymphocytes in the most melanoma patients are highly expressed the molecules of the VISTA [50]. In addition, high VISTA expression in the T cell tightly implies the worse survival in cancer patients. VISTA is highly expressed in many human cancers cell, therefore the clinical application of the anti-VISTA antibody significantly improves the overall survival of patients [51,52]. Other than this, in 2020, it has published that the development of a nanomolar anti-VISTA antibody that is expected to be used in the Phase I clinical trial [53].

#### 2.2.5. TIM-3

T-cell immunoglobulin and mucin domain-containing molecule 3 (Tim-3), belongs to the immunoglobulin superfamily and is a type | transmembrane glycoprotein. As a negative regulatory immune molecule, it is expressed on DC, CD4^+^T cells, CD8+T cells and several other cell types [54,55]. CEACAM1 have been acted as one of the important ligands of TIM-3 [55]. Meanwhile, observation indicates that exhausted T cell is characterized by high expression of IRs, in the meanwhile, including TIM-3. Interestingly, T cell exhaustion, but nit apoptosis, can be reversed by blocking immune checkpoints and restoring cytotoxic function [55]. Recently, according to the clinical datasets, some TIM-3 antagonistic mAbs have been entered on clinical trials. The typical example is that TSR-022 (Cobolimab) which are known as the a novel IgG4 anti-TIM-3 mAb entered the first phase I clinical trial in recent years and when combined with TSR-042 demonstrate favorable safety and efficacy [56,57].

Overall, many cell surface inhibitory receptors including PD-1, LAG3 and so on are happened in the coregulation of T cell exhaustion [58]. Researchers have shown diverse co-expression of multiple inhibitory receptors can damage on functionally tumor-specific cytotoxic T cells functionally, and the degree of functional defects in T cells correlates with the number and type of inhibitory receptors [18,31,59]. Therefore, it is strongly advised that the combination therapy with IRs inhibitors is the more feasible options in the clinical cancer treatments [56]. Currently, although there are many studies on inhibitory receptors, the detailed roles and regulatory mechanisms of the downstream pathways of different inhibitory receptors remain unclear. The molecular signaling pathways in exhausted T cells during LCMV infection or tumor progression are shown in Figure A2.

### 2.3. Unique Metabolic Patterns

Metabolic regulation consequently changes throughout the T cell cycle, from special initial naive state to the activation state after antigen presentation, and later exhaustion. Each phase of the T cell cycle has a different metabolic pattern and activity that either regulates or is regulated by related cellular signaling pathways, that profoundly influences cellular activation, function and survival [60,61].

#### 2.3.1. Naive T Cells

T cell can powerfully attack and resist pathogens and tumor cells [62,63,64]. Via flow cytometry, multiple subtypes of the T cell can be visualized, in combination with selective extracellular markers and intracellular transcription factors (TFs) [65]. The naïve T cells are characteristics of the homogenous antigen-inexperienced cells and with distinctive marker of CD45RA and CD62L. Therefore, it is consequently considered that naïve CD4^+^T cells and CD8^+^T cells (characterized by CD45RA^+^CCR7^+^CD62L^+^CD27^+^CD28^+^IL-7Rα^+^) [64,66]. Succeeding antigen encounter, naïve CD4^+^ T cells [66,67] constantly keep differentiate towards CD4^+^T_EFF_ which are constitute of various cell types including T helper 1 (Th1), Th17. however, CD8+T cells that encounter the specific antigen exert killing tumor ability by differentiating the subtypes of cytotoxic T cells.

Naive T cells require lower basal metabolic activity and energy intake and produce ATP in the comparison of the cytotoxic T cells, which are supported by the pathways of the oxygenation (oxidative phosphorylation, OXPHOS), directly contributing to T cell homeostasis [60,61,68].

#### 2.3.2. Effector T Cells

During the stage of the naïve T cells differentiate into T_EFF_ [63], T cells will definitely secrete large amount of the cytotoxic granules and the kind of cytokines to quickly induce cytotoxicity. T_EFF_ are the subtype of the cells which are basically negative for CD27 and CXCR4, nevertheless, they highly express markers of Killer cell lectin-like receptor subfamily G, member 1 (KLGR-1), the CCR10 [69] and so on and their markers are implied the terminal phase in the T cell development.

Undoubtedly, effector T cells cannot but change their metabolic patterns to meet the various needs of cell proliferation [70], differentiation [71,72], and effector function. These metabolic pathways are referred to the aerobic glycolysis, mitochondrial biogenesis, therefore accelerate the process of the respiratory chain phosphorylation (also known as OXPHOS) and one-carbon synthesis, correlating to mechanisms that target the PI3K/AKT/mTOR signaling pathway [73]. In T_EFF_, activation of the PI3K-AKT-mTOR pathway promotes aerobic glycolysis to satisfies its own energy needs [72]. Activation of the mTOR complex leads to the higher expression of the HIF-1α and c-Myc; these mainly upregulate the expression of enzymes including Glucose Transporter 1 (GLUT1) associated with glucose breakdown, thereby improving the T_EFF_ uptake of glucose and glutamine [74,75].

#### 2.3.3. Memory T Cells

Memory T cells, including stem cell memory T cells (Tscm), are highly expressed the many special molecular (typically CD95, IL-2Rβ, CXCR3), while central memory T cells (Tcm), which is characterized by the CD45RO^+^ CD28^+^CD62L^hi^CCR7^+^, establishes memory function and quickly acts to allow for rapid and accurate recall response to any future encounters with the same antigen or disease [69,70]. When encountering kinds of the tumors, memory T cells can migrate to the matching immune organs mainly including lymph nodes. Several studies have demonstrated that the in the TME, higher numbers of memory CD8+ T cells around the tumor are remarkably beneficial in the clinical patient positive prognosis [63,76].Memory T cell move from the aerobic glycolysis pathway to the use of mitochondrial fatty acid oxidation (FAO) to produce more ATP [77]. During the acute response and hypoxia status, such as growth factor deprivation, T cells may open the AMPK signaling pathway and correspondingly inhibit mTOR signaling. During anabolism shut-down, memory T cells switch to use glucose to synthesize lipids that undergo rapid LAL-mediated cell-intrinsic lipolysis to generate free fatty acids for FAO, demonstrating a degree of metabolic self-reliance [70,78].

#### 2.3.4. Exhausted T Cells

In the tumor patients, tumor cells firstly utilize aerobic glycolysis patterns, resulting in metabolic programming and conferring a survival advantage [79]. Likewise, exhausted CD8^+^T cells in the nutrient-and oxygen-deprived TME display “metabolic plasticity” during chronic tumor-associated antigen (TAA) stimulation [80], resulting in energy need shortage [81,82,83]. Several studies have indicated that CD8^+^T cells which are artificially expressed IRs mainly associate with metabolic disorders characterized by mitochondrial energy dysregulation [84,85]; it is therefore of great clinical significance to investigate the unique metabolic patterns of exhausted T-cells [85,86]. Current research focuses on the following specific mechanisms [85]; (1) metabolites of the tumor itself and (2) competition between various cells especially tumor cells and T cells in the TME for metabolic raw materials including oxygen and ATP. Classic mouse models have shown that PD-1 causes an increase in glucose in the TME, indicating prospective reversal of T-cell exhaustion and improved tumor clearance [87].

### 2.4. Specific Transcriptional and Epigenetic Features

#### 2.4.1. Transcriptional Changes in Exhausted T Cells

Specific transcriptional profiles are broadly changed in the exhausted T cells compared to state of the T_EFF_ or T_MEM_ CD8^+^T cells [86,87]; that is, specific TFs involved in various stages of T cell differentiation might serve as promising markers of T_EX_.

##### Tox

Tox is a focused TF that regulates T cell exhaustion in the recent studies [88]. Some studies have shown that Tox acts as a kind of the key factor during the differentiation and of exhausted T cells and may induce T cell exhaustion [89]. Exhausted T cells highly express Tox which can facilitate the expression of the PD-1, therefore promoting PD-1 translocation to the CD8^+^T cells membrane surfaces and helping the T cell exhaustion [90].

##### T-Bet and Eomes

T-bet and Eomes are highly homologous in effector T cell development [91]. During viral infection processes, the TF of T-bet and Eomes will become highly elevated with T-cell activation [91]. Both have a lot in common in regulating cell differentiation and function, moreover, they compete for Pdcd1 T-box region [91,92]. High nuclear T-bet strongly represses Pdcd1 gene expression in T_MEM_, while low nuclear T-bet in T_EX_ mainly repress of Pdcd1 in the lower levels [92,93]. During T cell differentiation, T-bet shows the highest expression level on effector T cells, whereas EOMES are inclined to subtypes of the memory T cells. The classification of the T_EX_ has been designated according to the expression of Eomes, T-bet, and PD-1 [94]. According to several studies, the T-bet^hi^Emoes^lo^PD-1^int^ subset can be re-activated by PD-1 pathway blockade, whereas the Eomes^hi^PD-1^hi^ subset is difficult to effectively respond to PD-1 blockade. These findings suggest that PD-1 pathway blockades resulted in T_EX_ epigenetic and transcriptional landscapes rewiring. It has been established that key TFs specifically binds to distinct genes upstream in various contexts in a specific time and space [95,96]; for example, during the acute period, the co-expressed TFs of Eomes were effector-biased genes. In contrast, the chronic neighbors of Eomes became progressively more upregulated over time and were involved in the immune response and CD8^+^T cell differentiation [95].

##### Tcf-1

T cell factor 1 (TCF-1) is a TF of the canonical Wnt signaling encoded by Tcf7 and orchestrates the Eomes-IL-2Rβ expression circuit [97,98]. Tcf-1 maintains the stemness capacity in T_EX_ cells and memory T cells. Mechanistically, Tcf-1 promote the TF of the Eomes expression and navigating c-Myc expression which regulates Bcl-2, helping to generate T_EX_ precursor cells [99,100].

Transcription of genes encoding molecules shown to be participated in the TCR-cytokine receptors signaling, also take part in the T cell exhaustion [101]. Moreover, additional TFs demonstrate be involved in T cell exhaustion, IRF4 [102], BATF [103,104], and NFATc1 [105] promote the expression of inhibitory receptors, including PD-1, and mediate impaired cellular metabolism. Furthermore, BATF interacts with IRF4 at AP-1–IRF consensus elements (AICEs) to modulate immune-regulatory networks [106,107].

#### 2.4.2. Epigenetic Landscape of T Cell Exhaustion

In mammalian cells, epigenetic mechanisms enable cells to differentiate, adapt to environmental changes, and propagate their cellular state after cell division. The epigenetic regulation information can be inherited across several generations in organisms [108,109]. Each gene sequence code keeps extremely conserved throughout many kinds of these cellular developments; however, epigenetic mechanisms can produce heritable phenotypic changes without a change in DNA sequence [110].

It is reported that exhausted T cells in cancer and chronic viral infections shows distinctive patterns of special related genes expression, including the sustained expression of PD-1. Insights from the epigenetic studies, there has revealed that T_EX_ cells are a unique lineage of cells that are epigenetically different from T_EFF_ and T_MEM_ cells, as B cells are from monocytes [111]. In the epigenetic landscape, T_EX_ cells contain approximately 6000 differentially accessible chromatin regions compared to T_EFF_ and T_MEM_ cells. Researchers have compared acutely activated T cells with terminal CD8^+^T_EX_ cells by biological database such as ATAC analysis and found that CD8^+^T_EX_ cells have many special properties, including *de novo* accessibility of the region at at-22.4 kb upstream of the murine Pdcd1 (PD-1) locus [111]. The epigenetic mechanisms regulating T-cell exhaustion are stable and inherited; for example, after PD-1 blockade, it is finest change is that there exists just almost ~10% of the epigenetic landscape was ‘re-activated’ to resemble the effector T-cell landscape implying that the reacquisition of the Tex strongly relates to chromatin state [112].

Overall, the transcriptional and epigenetic mechanisms of T_EX_ are complex and future in-depth studies on the regulatory mechanisms and factors of T-cell exhaustion are essential [16].

## 3. Factors Related to the Development of T Cell Exhaustion

### 3.1. Duration of Antigenic Stimulation

The duration of continuous antigen stimulation is the key contributors in T cell exhaustion in many chronic infection mouse models [113,114]. For instance, when CD8^+^T cells isolated from mice which are experiencing chronic LCMV/HBV (approximately one week) were transferred to immunodeficient mouse models, these cells would differentiate into normal memory T cells and ease T cell exhaustion [84]. However, if these T cells were exposed to persistent antigen for more than half of a month or even longer, the diminished functions of T cells would become irreversible. Researchers have shown that genes associated with TCR signaling, including EGR2 [115], EZH2 [116,117] IRF4 [118], and NR4A [119] are highly enriched in precursor exhausted CD8^+^T cells, mainly due to persistent antigen stimulation leading to T cell exhaustion. 

### 3.2. Soluble Cytokines

During chronic infection and cancer, the TME highly exists various pro-inflammatory factors, which greatly contribute to setting free complex negative regulatory factors, for instance IL-10, TGF-β, and IL-2 [120,121], thereby making T cell exhaustion appeared.

#### 3.2.1. IL-10

IL-10, a cytokine closely associated with attenuated T-cell activation, which induces STAT-3 signaling, is upregulated in persistent infections with several viruses such as LCMV, HBV, and EBV. In the related mouse model, blockade of IL-10 in combination with the immune-checkpoint inhibitors targeting PD-1 or PD-L1 [122] enhances the tumor-killing capacity of T_EFF_ and particularly develops the response of exhausted T cells, there has been suggested that IL-10 [123] make a specific and indispensable role in the development of T cell exhaustion.

#### 3.2.2. TGF-β

TGF-β [124] also be tightly linked to the promotion of T-cell exhaustion. It also regulates immune response and maintains immune homeostasis, mainly by the way of regulating proliferation and differentiation of various immune cells. TGF-β participates in the development of the T-cell exhaustion and inhibits TCR signaling transduction pathways by regulating the downstream expression of SMAD2 [125,126].

#### 3.2.3. IL-2

Apart from IL-10/TGF-β, IL-2 also plays a key driver of the of the formation of the CD8+T cell exhaustion. It is often thought that IL-2 is a kind of the cytokines that also known as T cell growth factor essential for the proliferation of T cells [120,121]. Indeed, at the early course of tumor growth, IL-2-mediated induction of BLIMP1 expression is required for effector CD8^+^T cells growth and differentiation. However, in the TME, excessive IL-2 acts on CD8^+^T cells to promote their exhaustion in the later stages of continuous tumor growth. Exhausted CD8^+^T cells express higher levels of IL-2Rβ, which are co-expressed the T cell exhaustion marker including PD-1 and TIM3 and so on, therefore loss or downregulation of IL-2Rβ tends to help to reverse the exhausted phenotypes [127]. A series of studies targeting IL-2 found that IL-2 signal can activates STAT5, induces CD8^+^T cells to synthesize large amounts of TPH1 (Tryptophan Hydroxylase 1), which produces 5-hydroxytryptophan (5HTP) that subsequently binds to the arylhydrocarbon receptor (AhR), hence promoting AhR entry into the nucleus, and its direct upregulation of immunosuppressive receptors, thus instigating therapy inducing CD8+T cell exhaustion [120].

Therefore, IL-2 acts as the double-edged sword, although IL-2 signaling can promote an exhausted phenotype, it is also a feasible pathway to reverse CD8^+^ T-cell exhaustion. Initially, the earliest therapeutic applications of IL-2 were to improve immune responses in cancer patients [128,129]. In present, combining IL-2 treatment with immune checkpoints inhibitors PD-1 has striking synergistic effects for re-invigorating exhausted CD8^+^T cells [130].

#### 3.2.4. IL-21

IL-21 is mostly produced by CD4^+^T cells and has been shown to play a central role in the maintenance of CD8^+^T cell cytotoxicity [131]. Acute and chronic infections and pathogen invasion result in differing levels of IL-21 production change in the complex TME, which directly influences the generation of CD8^+^T cells [132]. Additionally, IL-21 directly prompts the expression of BATF which has been shown to be predominantly expressed in B and T cells [133], which are mainly involved in sustaining antiviral CD8 T_EFF_ cytotoxicity [134] and in some cases, fostering T cells exhaustion. Thus, IL-21 may maintain CD8^+^T cell responses and at least partially oppose T cell exhaustion [135].Some studies have shown that targeting IL-21 signaling pathway in the tumor-reactive T cells can greatly promote the generation of memory stem T cells (T_SCM_) with enhanced cell proliferation [136,137].

In summary, the development and inducing factors in T-cell exhaustion is a complex outcome driven by various factors, such as secreted cytokines, which further promote T-cell exhaustion, therefore make the continuous deterioration and disease progression appeared. Although a great deal has been done to aim at the effects of T-cell exhaustion, related specific mechanisms and various complex interactions of each factor remain a gap that requires further investigation. These concepts are discussed in further detail below and are illustrated in Figure A3.

## 4. Reversion of the T Cell Exhaustion

Exhausted T cells manifest as the functionally defective state that makes the body tolerant to tumor effects and it is almost impossible to completely reverse the exhaustion state. However, more recent studies have reported that IL-2 has the potential to reverse T cell exhaustion. And the underlying mechanism is that, IL-2 signals can change their differentiation direction on precursors of exhausted CD8^+^T cells (PD-1^+^TCF1^+^ stem-like CD8^+^T cells) [138]. Many clinical experimental researches in mice have seen that, to some extent, blocking inhibitory receptors and inhibitory cytokines may be a key factor to reverse T cell exhaustion and enhance anti-tumor immunity outcome in the cancer patients especially terminal.

The clinical application of blocking PD-1 with anti-PD-1 antibodies displays successful outcomes, and it is reported that blockade of PD-1 or PD-L1 improves the cytolytic activity of exhausted CD8^+^T cells, enhances inflammatory cytokines production, and reduces the viral load and tumor burden in a mouse model [14,31,43]; this implies that the exhausted T cell state is not irreversible. Preclinical studies have demonstrated in various animal tumor models, that anti-PD-1 immune checkpoint inhibitors can effectively alter T cell exhaustion, and even CD8^+^T cells become more capable of fighting tumors. For instance, in a mouse melanoma model, anti-PD-1 antibodies blocked the rate of hematogenous metastasis of B16 melanoma cells [122,139].

In addition, researchers found that blocking PD-1 in combination with other immune checkpoint inhibitors was more effective than blocking PD-1 alone [140,141]. Immune checkpoint inhibitors present a pathway to fight tumors; however, drug resistance remains one of the key limitations. Using combinations of immune checkpoint inhibitors may improve or eliminate resistance. In animal models of anti-PD-1 resistance, blocking the negative regulatory pathway of TIM-3 aids in reversing the effects of anti-PD-1 resistance, and therefore of T-cell exhaustion [33,59].

In the TME, various cytokines are also involved in T cell exhaustion hence, regulating cytokines to reverse T cell exhaustion presents an effective therapeutic strategy. Previous studies have found that the combination of IL-2 and antibodies against PD-1 was effective in eliminating a large tumor burden and reducing viral load in a mouse model [138,142]. Similarly, blockade of IL-10 signaling may effectively improve T-cell exhaustion as well as control persistent virus infection [143].

In the TME, tumor cells preferentially change the specific metabolic patterns such as aerobic glycolysis to their survival. Targeting metabolic reprogramming of TME to reinvigorate the states of exhausted CD8+T cells is a superior choice [144]. Some studies shows that 2-DG, an inhibitor of glycolysis, was utilized to promote the production of memory cells, resulting in enhanced metastatic cells in tumor-bearing mice persistence and function [145]. Inhibiting glycolytic metabolism enhances CD8+ T cell memory and antitumor function. Similarly, inhibiting AKT signaling in vitro promotes memory-enhancing metabolic profile of CD8+T cell generation [83,114]. All in all, metabolic and immune signaling are highly correlated, combining metabolic interventions with immunotherapy to restore anti-tumor activity of exhausted T cells that is highly likely to improve response rate to immunotherapy [144].

## 5. Prospect

When it comes to tumor immunity, T_EX_ cells, a unique type of T cell, become a major barrier, therefore “re-invigorating” T_EX_ cells provide opportunities to improve anti-tumor immunotherapy. Nowadays, the mechanism of T-cell exhaustion has been studied in depth, and successful clinical cases of the reversal of T_EX_ cells for immunotherapy have become increasingly common. However, only some patients have achieved complete remission of the disease, and serious side effects, including cytokine storms, have occurred. Therefore, the mechanisms of T_EX_ need to be further investigated, and reversing T-cell exhaustion for clinical application requires comprehensive study to provide a sufficient theoretical basis for subsequent clinical tumor immunotherapy. With the deepening and comprehensive researches, we believe that the molecular mechanisms underlying T cell exhaustion will be made clearer.

## 6. Summary

In the clinical aspects, cancer causes millions of deaths each year and is one of humanity’s greatest health challenges. After the 20th century, it is appeared that a new principle of cancer treatment by activating the body’s own immune system’s ability to attack tumor cells, and tumor immunotherapy has achieved good success in clinical practice. However, with the continuous researches on the T cells in the TME during the immune response against cancer, a new concept of “T cell exhaustion” was created. T cell exhaustion is tightly correlated with poor prognosis of the clinical cancer and viral infection treatment. Thus, at present, there is doubt that exploring regulatory mechanisms and reversing of CD8^+^T cell exhaustion is the more wise and valid pathway in the cancer treatment. We follow the emerging researches trends and provide a systematic review on the T cell exhaustion especially the factors and clinical applications in the T cell exhaustion. At the same time, we also recognize that we may be inadequately aware of the T cell exhaustion and require more updated contents.

## Data Availability

Not applicable.

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
