# Peer review of "Regulatory Mechanisms and Reversal of CD8+T Cell Exhaustion: A Literature Review"

_biology, 2023, doi:10.3390/biology12040541_

Round 1
Reviewer 1 Report
Please remove duplicate reference 137 0r 138.
Author Response
Dear reviewer, First of all, thank you very much for your precious time to read my article and put forward valuable comments and opinions. To your questions, I tried my best to reply one by one, hope this is helpful for you.
- Reference 137 has been removed.

Reviewer 2 Report
the topic of the paper is potentially interesting
However, huge changes are necessary for the paper to even be considered for publication, as the paper is flawed in almost all areas
1. In the introductory part, it should be emphasized that in the tumor and stroma system, in addition to T cells, NK cells and NKT cells play a major role.
It has been shown that these subpopulations of cells have a reduced role in cytokine production as well as a reduction in receptors for activation
a) Decreased Interferon γ Production in CD3+ and CD3- CD56+ Lymphocyte Subsets in Metastatic Regional Lymph Nodes of Melanoma Patients. Pathol Oncol Res. 2015 Sep;21(4):1109-14.
b) Distribution of several activating and inhibitory receptors on CD3(-)CD56(+) NK cells in regional lymph nodes of melanoma patients. J Surg Res. 2013 Aug;183(2):860-8.
After that, at the end of the introductory part, it was said that the aim of the work was to examine the function of T lymphocytes, although there are already data for such a dysfunction.
2. In addition, the authors did not show the importance of other molecules
add references and comment on these markers:
Decreased expression of pSTAT, IRF-1 and DAP10 signaling molecules in peripheral blood lymphocytes of patients with metastatic melanoma. J Clin Pathol. 2016 Apr;69(4):300-6.
3. at the end of section 2.1. Progressive Loss of Effector Function add references that show that during tumor progression the decrease in the function of cells of the immune system is a consequence of the production of cytokines from the tumor that inhibit cells as shown previously and add a reference at the end of the paragraph: a) Multiomic analysis of cytokines in immuno-oncology . Expert Rev Proteomics. 2020 Sep;17(9):663-674.
b) The role of cytokines in the regulation of NK cells in the tumor environment. Cytokines. 2019 May;117:30-40.
4. in the Naive T cells section, add phenotypic characteristics and the way of defining this population of cells.
5. in section 2.3.2. Effector T cells are not the most important role of metabolism, but it is necessary to say how these cells perform their function. Therefore, these populations are not called effectors because they have a disturbed metabolism and secrete mediators. By the way, the metabolic function is disturbed in them, and this section should be completely changed or called a metabolic disorder, and before that the function of effector T cells should be clearly defined.
6. during the progression of the tumor, T cells show damage to the metabolism as well as to the membrane, and there is a release of intracellular enzymes, which is very important and to add references that showed this a long time ago and were not cited here. : TNF-alpha induces changes in LDH isotype profile following triggering of apoptosis in PBL of non-Hodgkin's lymphomas.
Ann Hematol. 2004 Feb;83(2):84-91.
7. in the memory T cell section, the immunophenotype and main characteristics as well as immunophenotype changes during cancer progression are not written, which is very important
8. in section 2.4.1. Transcriptional changes in exhausted T cells add to the immunophenotype of these cells
9. in section 3.2.2. TGF-β add references Increased circulating TGF-β1 is associated with impairment in NK cell effector functions in metastatic melanoma patients. Growth Factors. 2022 Nov;40(5-6):231-239.
10. section 3.2.3. IL-2 needs a complete change.
This story for IL2 and others as well as the references given in the attachment refer to viral infections and not to tumors. Therefore, only some tumors are the result of a viral infection. Change the text and correct the references for IL2 as well as for changes in the IL2 receptor, which is more adaptive and not associating IL-2 with dead and ineffective cells, and . add appropriate references. by the way The first drug to enhance immunity was IL-2 since 1990, and IL-2 was described in the protocols as immunostimulatory. So, the facts that are presented here are not exactly supported by evidence, so change it.
Author Response
Dear reviewer, First of all, thank you very much for your precious time to read my article and put forward valuable comments and opinions. To your questions, I tried my best to reply one by one, hope this is helpful for you.
Point 1: In the introductory part, it should be emphasized that in the tumor and stroma system, in addition to T cells, NK cells and NKT cells play a major role. It has been shown that these subpopulations of cells have a reduced role in cytokine production as well as a reduction in receptors for activation.
Response 1: I can't agree with you more. At the end of the introduction, I mainly introducing NK cells which reduce role in cytokine production such as interferon-γ(IFNγ), regulates adaptive antitumor response. And citing the relevant references you proposed.
Point 2: In addition, the authors did not show the importance of other molecules, add references and comment on these markers: Decreased expression of p-STAT, IRF-1 and DAP10 signaling molecules in peripheral blood lymphocytes of patients with metastatic melanoma. J Clin Pathol. 2016 Apr;69(4):300-6.
Response 2: Peripheral blood lymphocytes (PBLs) typically exist anti-tumor function which are regulated by the numerous signaling molecules. I mainly introduce common cell surface molecules such as PD-1、LAG-3 and Tim-3,etc,. Based on your suggestion, I have added some signaling molecule to the article ate the beginning of the section 2.2.
Point 3: at the end of section 2.1. Progressive Loss of Effector Function add references that show that during tumor progression the decrease in the function of cells of the immune system is a consequence of the production of cytokines from the tumor that inhibit cells as shown previously and add a reference at the end of the paragraph:
- a) Multiomic analysis of cytokines in immuno-oncology . Expert Rev Proteomics. 2020 Sep;17(9):663-674;
- b) The role of cytokines in the regulation of NK cells in the tumor environment. Cytokines. 2019 May;117:30-40
Response 3: Completed, add the recommended reference at the end of section 2.1.
Point 4: in the Naive T cells section, add phenotypic characteristics and the way of defining this population of cells.
Response 4: The T cell compartment appear in different subsets, as the part of the naïve T cells, I added the classification of naïve T cells, and mainly differentiation of various antigens encountered. We evaluate the kinds and populations of cells mainly by flow cytometry.
Point 5: in section 2.3.2. Effector T cells are not the most important role of metabolism, but it is necessary to say how these cells perform their function. Therefore, these populations are not called effectors because they have a disturbed metabolism and secrete mediators. By the way, the metabolic function is disturbed in them, and this section should be completely changed or called a metabolic disorder, and before that the function of effector T cells should be clearly defined.
Response 5: I quite agree with your advice, I will further refine the characteristics of T cell exhaustion by associating effector cells with metabolic reprogramming.
Point 6: during the progression of the tumor, T cells show damage to the metabolism as well as to the membrane, and there is a release of intracellular enzymes, which is very important and to add references that showed this a long time ago and were not cited here. : TNF-alpha induces changes in LDH isotype profile following triggering of apoptosis in PBL of non-Hodgkin's lymphomas. Ann Hematol. 2004 Feb;83(2):84-91.
Response 6: Thank you for your recommendation, this reference is a basic plus clinical synthesis, I also think it is very necessary to add it to my article.
Point 7:in the memory T cell section, the immunophenotype and main characteristics as well as immunophenotype changes during cancer progression are not written, which is very important.
Response 7: I have been added some contents you suggested including the immunophenotype and main characteristics as well as immunophenotype changes during cancer progression.
Point 8: in section 2.4.1. Transcriptional changes in exhausted T cells add to the immunophenotype of these cells
Response 8: Have completed the addition on the immunophenotype of the exhausted T cells.
Point 9: in section 3.2.2. TGF-β add references Increased circulating TGF-β1 is associated with impairment in NK cell effector functions in metastatic melanoma patients. Growth Factors. 2022 Nov;40(5-6):231-239.
Response 9: This is a relatively new reference which published just last year. Have supplemented the specific reference on the section 3.2.2.
Point 10: section 3.2.3. IL-2 needs a complete change
Response 10: Here, I have added IL-2 how to induce T cell exhaustion and current clinical application and drug discovery for immunotherapy. And minor changes in correct the cited appropriate references.

Reviewer 3 Report
See the attached document.

Author Response
Dear reviewer, First of all, thank you very much for your precious time to read my article and put forward valuable comments and opinions. To your questions, I tried my best to reply one by one, hope this is helpful for you.
Responese 1:
In response to your opinion, I have made the following modifications. First, I have made a table which is graphically depicted that chronic antigen stimulation facilitates different stages of T cell exhaustion to summarize and conclude according to the content of the article, so as to make readers more clear. I also corrected the all minor changes and details.
Response 2:
- Title: Regulatory mechanisms and reversal of CD8+T cell exhaustion: A systematic review.
- Abstract: add more data to make T cells exhaustion clearly.
- Keyword: complete replacement “tumour immunotherapy” with “immunotherapy”.
- Change the error”CTLA-1” to “CTLA-4”.

Reviewer 4 Report
The authors of the present article "Regulatory mechanisms and reversal of CD8+T cell exhaustion" successfully and comprehensible describe T cell exhaustion in different disease scenarios and ways to revert it. The article is well structured and well written. However, some sections require some changes or further discussion.
1. Introduction:
Page 1Line 19: did the authors mean that exhaustion was first described in that context of HIV infections?
2. The Features of T cell Exhaustion
Page 2 line 21: Please describe the acronym ITIM. Since being an ITIM IR is mentioned for each of the IRs described in the text, please describe how is it relevant that and IR is ITIM or non-ITIM?
Page 3 line 27: The study in reference 43 is not relevant to TIGIT, since it talks about TIM3. Please add a separate section describing the role of TIM3 in T cell exhaustion and another one for CTLA-4. Or a pool section talking about other IRs including TIM3, CTLA-4, BTLA, etc.
2.3. Unique metabolic patterns
This whole section is very interesting, but it would be more scientifically relevant that the metabolic state of the naive, effector and memory T cells was integrated with and discussed in opposition to the exhausted T cells.
2.4.2. Epigenetic landscape of T cell exhaustion
Page 6 line 17: reference 117 is the same as 116. Moreover, in this section this reference is worth mentioning: https://pubmed.ncbi.nlm.nih.gov/34788079/
3.2.3. IL-2
IL-2 is a T cell growth factor, it is essential for the proliferation and survival of T cells and the formation of effector and memory cells. Please acknowledge that in the discussion of how it also plays a role in exhaustion.
4. Reversion of the T cell exhaustion
It would be relevant to discuss the differences between exhausted T cells and terminally exhausted T cells. Moreover, move some of the discussion on clinical trials in section 2.2. Sustained and High Expression of Inhibitory Receptors, to this section.
Are there any studies targeting metabolic, transcriptional or epigenetic factors leading to T cells exhaustion? Please discuss them in this section.
Author Response
Dear reviewer, first of all, thank you very much for your precious time to read my article and put forward valuable comments and opinions. To your questions, I tried my best to reply one by one, hope this is helpful for you.
Point 1: Introduction: Page 1Line 19: did the authors mean that exhaustion was first described in that context of HIV infections?
Response 1: sorry, I don’t explain it clearly and that's not the understanding. I want to explain the definition of T cell exhaustion has been used to describe the response of the T cells to chronic viral infection, in the context of persistent HIV infection, not the first described in that context of HIV infections.
Point 2: Page 2 line 21: Please describe the acronym ITIM. Since being an ITIM IR is mentioned for each of the IRs described in the text, please describe how is it relevant that and IR is ITIM or non-ITIM?
Response 2: ITIM, immunoreceptor tyrosine-based inhibitory motif, mainly exists in the intracellular part of these some IR molecules. Not all IRs contain the ITIM. For example, LAG-3 functions as a negative regulator via the specific KLEELE motif.
Point 3: Page 3 line 27: The study in reference 43 is not relevant to TIGIT, since it talks about TIM3. Please add a separate section describing the role of TIM3 in T cell exhaustion and another one for CTLA-4. Or a pool section talking about other IRs including TIM3, CTLA-4, BTLA, etc.
Response 3: For this problem, I make adjustments in reference 43 and add a separate section describing the role of TIM3 in T cell exhaustion.
Point 4: This whole section is very interesting, but it would be more scientifically relevant that the metabolic state of the naive, effector and memory T cells was integrated with and discussed in opposition to the exhausted T cells.
Response 4: Considering the metabolism of the T cell exhaustion, I systematically add some extra content.
Point 5: Page 6 line 17: reference 117 is the same as 116. Moreover, in this section this reference is worth mentioning: https://pubmed.ncbi.nlm.nih.gov/34788079/
Response 5: As for the repeated references, I have deleted one of them and quoted the key references you recommended in this section.
Point 6: IL-2 is a T cell growth factor; it is essential for the proliferation and survival of T cells and the formation of effector and memory cells. Please acknowledge that in the discussion of how it also plays a role in exhaustion.
Response 6: I have already linked IL-2 to PD-1 expressed on the surface of T cells, but its role in T cell exhaustion is not complete, and I have added the relevant content, such as IL-2 signal induce CD8+T cell exhaustion through metabolic reprogramming.
Point 7: It would be relevant to discuss the differences between exhausted T cells and terminally exhausted T cells. Moreover, move some of the discussion on clinical trials in section 2.2. Sustained and High Expression of Inhibitory Receptors, to this section. Are there any studies targeting metabolic, transcriptional or epigenetic factors leading to T cells exhaustion? Please discuss them in this section.
Response 7: The difference between exhausted t cells and terminal exhausted T cells is mainly due to the different expression of transcription factors especially EOMES and T-bet, which was discussed earlier in the 2.4.1.2 section. I again complemented the related contents to make the T cells exhaustion clearly.

Round 2
Reviewer 3 Report
See the attached document.

Author Response
Dear reviewer, First of all, thank you very much for your precious time to read my article and put forward valuable comments and opinions. To your questions, I tried my best to reply one by one, hope this is helpful for you.
Responese 1: I have changed the title according to your opinion and added a relevant figure according to the section 2 parts.
